# Eigenstate thermalization to non-Gibbs states in strongly-interacting chaotic lattice gases

Vladimir A. Yurovsky

*School of Chemistry, Tel Aviv University, 6997801 Tel Aviv, Israel*

Amichay Vardi

*Department of Chemistry, Ben-Gurion University of the Negev, Beer-Sheva 84105, Israel and*
*ITAMP, Harvard-Smithsonian Center for Astrophysics, Cambridge, MA 02138, USA*

We demonstrate that equilibrium energy distributions in many-body chaotic systems can be qualitatively different from the Fermi-Dirac and Bose-Einstein distributions. This effect can appear in systems with finite energy spectra, which support both positive and negative temperatures, in the regime of quantum degeneracy, when the eigenstates fill a substantial part of the Hilbert space. The results are supported by exact diagonalization calculations for chaotic Fermi-Hubbard and Bose-Hubbard models, when they have Wigner-Dyson statistics of energy spectra and demonstrate eigenstate thermalization. The proposed effects may be observed in experiments with cold atoms in optical lattices.

The properties of complex systems in thermodynamic equilibrium are determined by a few thermodynamic parameters, such as temperature, pressure, and density. Chaotic systems relax to equilibrium independently of their specific initial state. However, an isolated quantum system is described by the Schrödinger equation and, having been created in one of its eigenstates, will remain in that state forever.

This paradox is resolved by the eigenstate thermalization hypothesis (ETH) [1, 2] (see also [3, 4], the experimental work [5], the review [6] and the references therein). It states that the vast majority of a chaotic system's eigenstates behave as statistical ensembles. Consequently, the expectation value of any local observable $\hat{O}$, evaluated for *any* eigenstate $|\alpha\rangle$ of a chaotic system, is approximately equal to its microcanonical mean over the pertinent energy shell,

$$\left\langle \alpha \left| \hat{O} \right| \alpha \right\rangle \approx \overline{\left\langle \alpha \left| \hat{O} \right| \alpha \right\rangle} \equiv \frac{\Delta_\alpha}{\Delta_{\mathrm{MC}}} \sum_{\alpha' \in \mathrm{MC}(E_\alpha)} \left\langle \alpha' \left| \hat{O} \right| \alpha' \right\rangle,$$

(1)

where $E_\alpha$ is the eigenstate energy, $\alpha \in \mathrm{MC}(E)$ means that $|E_\alpha - E| < \Delta_{MC}/2$, $\Delta_{\mathrm{MC}}$ is the microcanonical shell width, and $\Delta_\alpha$ is the average distance between the adjacent $E_{\alpha'}$ in the vicinity of $E_\alpha$.

Equation (1) provides an equilibrium state that is independent of the initial state details, but does not provide the equilibrium state properties. For a low-density gas of interacting particles in a flat potential the equilibrium state agrees with the microcanonical ensemble for an ideal gas, as proven in [2] on the basis of the Berry conjecture [7]. In the thermodynamic limit, where the number of particles and the system's volume are increased while keeping a fixed particle density, the microcanonical ensemble is equivalent to the canonical one. In this case, the Gibbs (Fermi-Dirac or Bose-Einstein) momentum distributions are obtained for the respective permutation symmetry, with the standard relation be-

tween the temperature and the total gas energy, which is equal to the eigenstate energy. Such distributions were also obtained for Fermi [8, 9] and Bose [10] systems close to quantum degeneracy, but now the temperatures are shifted. ETH means that the eigenstate to eigenstate fluctuations of expectation values within any chaotic microcanonical shell are suppressed. In certain situations [11, 12], the fluctuation variance is inversely-proportional to the number of principal components (NPC) $\mathcal{N}_{\mathrm{PC}}$ — the estimate of the number of integrable system eigenstates comprising the non-integrable one. Thus, ETH typically implies large NPC, but can be practically attained when NPC is substantially smaller than the dimension $\mathcal{N}_{\mathrm{HS}}$ of the Hilbert space (which can be also constrained due to possible conservation laws). NPC approaches a large integer fraction (1/3 for time-reversible systems [13] or 1/2 for time-irreversible ones [14]) of $\mathcal{N}_{\mathrm{HS}}$ only in the regime of quantum ergodicity [15]. In this work we demonstrate quantitative deviations from the Gibbs distributions in certain strongly-interacting systems, particularly, in the regime of quantum ergodicity. These deviations can not be reduced to a mere temperature change.

Other reasons exist for the distribution deviations from the Gibbs ones. Integrable (and nearly integrable) systems keep complete memory of their initial state. Such effects have been observed in experiments with quantum Newton cradles [16, 17] and cold-atom breathers [18, 19]. Final states of relaxation (and distributions) for integrable systems are described by the generalized Gibbs ensemble [20] that accounts for the additional constraints imposed by the integrals of motion. Incompletely chaotic systems [21, 22] with a small number of degrees of freedom keep certain memory of their initial states. In many-body systems, eigenstate thermalization can also be prevented by many-body localization (MBL) [15, 23], vanishing in the thermodynamic limit (see also, e.g., [24–

27]). Even if eigenstate thermalization takes place, the distributions can deviate from the Gibbs ones due to moderate numbers of degrees of freedom in mesoscopic systems [28]. This effect, however, vanishes in large systems, unlike the one considered here.

We find eigenstates of two lattice models by direct numerical diagonalization, allowed up to the Hilbert space dimension $\mathcal{N}_{HS} \lesssim 2 \times 10^4$. Throughout this manuscript, all energies are measured in units of the lattice hopping energy. In the first, two-dimensional (2D) Fermi-Hubbard (FH) model, $N$ spin-polarized fermions have nearest-neighbor interactions with the strengths $V$ [29]. This model includes hoppings between the site $(l_x, l_y)$ and 8 neighboring sites: $(l_x \pm 1, l_y)$, $(l_x, l_y \pm 1)$, and $(l_x \pm 1, l_y \pm 1)$, where $l_x$ and $l_y$ label sites of the $L_x \times L_y$ lattice. Inclusion of hoppings with simultaneous change of $l_x$ and $l_y$, together with twisted-periodic boundary conditions, allow us to remove degeneracies of the many-body non-interacting particle eigenstates. The total number of one-body (1B) states in this model is $L = L_x L_y$. Due to the spatial homogeneity of this model, we consider separately each sector with the given total momentum which contains $\mathcal{N}_{HS} \approx (L-1)!/(N!(L-N)!)$ eigenstates. The results below are obtained for $N = 6$ particles in the $6 \times 5$ lattice ($L = 30$) and the total momentum $x$ and $y$ components 3 and 2, respectively. In this case, $\mathcal{N}_{HS} = 19811$.

The second model is a one-dimensional (1D) Bose-Hubbard (BH) chain with $N$ spinless bosons in $L$ sites, with on-site interactions of strength $V$ and hard wall boundaries [29]. Parity symmetry is broken by adding a random disorder/bias potential of order $\leq 0.05$. The resulting Hilbert space dimension for the bosonic system is $\mathcal{N}_{HS} = (N + L - 1)!/(N!(L-1)!)$. The system with $N = 10$ particles in $L = 8$ sites, considered here, has $\mathcal{N}_{HS} = 19448$.

Analyzing the chaotic system properties, we have to compare them to ones of the closest integrable system. For this purpose, we use corresponding systems of non-interacting particles. Their symmetric or anti-symmetric many-body eigenfunctions — the orbital Fock states $|n\rangle = |n_1 \dots n_L\rangle$ — have the eigenenergies $E_n = \sum_k n_k \varepsilon_k$. Here $n_k$ are occupations of the 1B orbitals, labeled in increasing order of their eigenenergies $\varepsilon_k$. Subtractions the average expectation values of interactions from the interacting particle Hamiltonians [29] leads to a substantial overlap of the non-interacting and interacting spectra $\{E_n\}$ and $\{E_\alpha\}$ .

For sufficiently strong interaction, both models become chaotic. For the FH model at $V = 1$ the ratio of two consecutive level spacings [30], averaged over the energy spectrum [29] increases to $\langle r \rangle \approx 0.525$ (cf. $\langle r \rangle \approx 0.536$ [31] for the Wigner-Dyson ensemble of Gaussian orthogonal matrices, describing completely-chaotic systems). The chaotic behavior is confirmed also by suppression of eigenstate-to-eigenstate fluctuations of the observable expectation values [29]. Their variances are reduced by

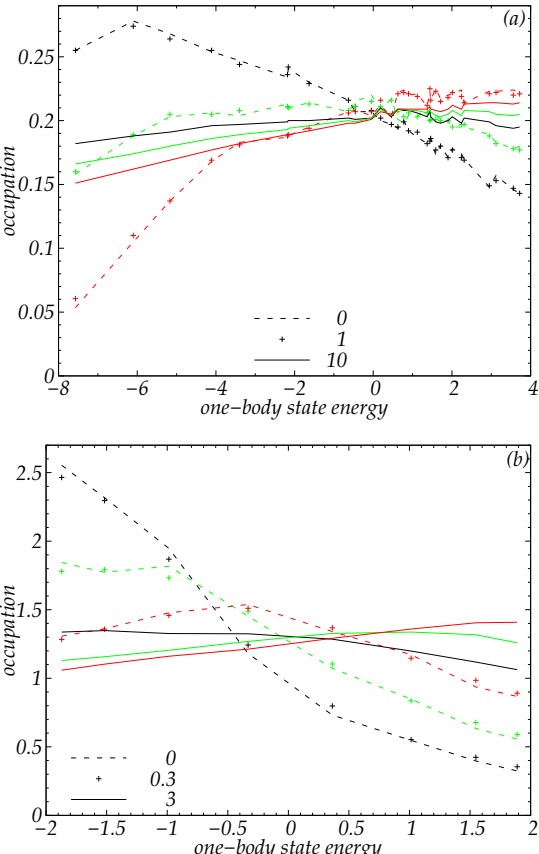

FIG. 1. 1B orbital occupations for (a) the FH model with the interaction strengths $V = 0$ (dashed lines), $V = 1$ (pluses), and $V = 10$ (solid lines), averaged over microcanonical shells with the mean energies -3 (black), 0 (green), and 3 (red) and (b) the 1D BH model with the interaction strengths $V = 0$ (dashed lines), $V = 0.3$(pluses), and $V = 3$ (solid lines), averaged over microcanonical shells with the mean energies $-8.57$ (black), $-5.14$ (green), and $-1.17$ (red).

two orders of magnitude. Another criterion of chaoticity, NPC, increased to $5 \times 10^2$ [29]. The caoticity of the BH model is determined by the value of $VN$. For the BH model at $VN = 3.0$ ($V = 0.3$) we have $\langle r \rangle \approx 0.529$, fluctuation variances are reduced by two orders of magnitude, and NPC is increased to $8 \times 10^2$. For these interaction strengths, the microcanonical distributions of the 1B orbital occupations for interacting and non-interacting particles are very close (see Fig. 1), both being different from the Gibbs distribution due to the small system size [28]. Due to macroscopic self-trapping, the BH model becomes integrable again at large $V$ where the site populations become effective integrals of motion. Thus, the chaoticity parameter reduces to $\langle r \rangle \approx 0.41$ at $V = 10$ (cf $\langle r \rangle \approx 0.386$ for integrable systems).

By contrast, when the interaction is increased (but remains in the chaos region for the BH model) the microcanonical distributions for interacting particles devi-

ate substantially from the non-interacting ones. This is shown in Fig. 1 for the FH model with $V = 10$, when $r \approx 0.53$ and NPC increases to the value of $6 \times 10^3$, about one third of $\mathcal{N}_{\mathrm{HS}} = 19811$ and for the BH model with $V = 3$, $r \approx 0.5$, and NPC $4.5 \times 10^3$ (about one quarter of $\mathcal{N}_{\mathrm{HS}} = 19448$). The difference between the distributions for interacting and non-interacting particles is qualitative and can not be reduced to a total energy shift, as evident from comparison with non-interacting particle distributions at different energies. As the eigenstate-to-eigenstate fluctuations are suppressed, the distributions for individual eigenstates deviate too.

The effect can be explained in the following way. Consider an observable $\hat{O}$ that commutes with the Hamiltonian of non-interacting particles, such that $\hat{O}\,|n\rangle = O_n\,|n\rangle$. The microcanonical mean (1) of its expectation value evaluated for eigenstates of interacting particles can be expressed as:

$$\overline{\left\langle \alpha \left| \hat{O} \right| \alpha \right\rangle} = \sum_n \Delta_\alpha W(E, E_n) O_n \qquad (2)$$

in terms of the local density of states (LDOS), or strength function [13]

$$W(E, E_n) = \frac{1}{\Delta_{\mathrm{MC}}} \sum_{\alpha \in \mathrm{MC}(E)} |\langle \alpha | n \rangle|^2 \qquad (3)$$

[see (1)]. The LDOS is generally a flat function of energies. If its energy span $\Gamma$ substantially exceeds $\Delta_{\mathrm{MC}}$, $O_n$ in (2) is effectively averaged and can be approximated by its microcanonical mean

$$\overline{O}(E) = \frac{\Delta_n}{\Delta_{\mathrm{MC}}} \sum_{n \in \mathrm{MC}(E)} O_n,$$

where $\Delta_n$ is the average distance between neighboring $E_n$ in the vicinity of $E$. Further, as $\Gamma$ substantially exceeds $\Delta_n$, approximating summation in (2) by integration, we get

$$\overline{\left\langle \alpha \left| \hat{O} \right| \alpha \right\rangle} \approx \int_{E_{\min}}^{E_{\max}} \frac{dE'}{\Delta_n} \Delta_\alpha W(E, E') \overline{O}(E'), \qquad (4)$$

where $E_{\min}$ and $E_{\max}$ define the support of the non-interacting system's spectrum $E_{\{n\}}$.

Consider a particular case of the orbital occupation operator $\hat{N}_k |n\rangle = \sum_{j=1}^{N} n_{k_j} |n\rangle$. The shape of the microcanonical distribution of the orbital occupations $\overline{N_k(E)}$ alters with the mean shell energy (see Fig. 1). If $W(E - E')$ vanishes when $|E - E'| > \Gamma$ and $\Gamma$ is small with respect to the energy scale on which the microcanonical distribution varies, we have $\overline{\left\langle \alpha \left| \hat{N}_k \right| \alpha \right\rangle} \equiv \overline{N_k^{int}(E)} \approx \overline{N_k(E)}$, thus justifying the equivalence between the occupation statistics of the interacting and non-interacting

systems. However, if $\Gamma$ exceeds this scale, the interacting-system's occupation distribution $\overline{N_k^{int}(E)}$ can mix non-interacting distributions $\overline{N_k(E)}$ of different shape and be different from any individual non-interacting microcanonical distribution $\overline{N_k(E)}$, as demonstrated by Fig. 1.

The exact diagonalization method is applicable only to small numbers of particles and lattice sites when the microcanonical distribution of the orbital occupations $\overline{N_k(E)}$ is different from the canonical distributions [28]. However, for large numbers of particles the microcanonical occupation of the orbitals $\overline{N_k(E)}$ are precisely given by the Gibbs (Bose or Fermi) distributions

$$\overline{N_k(E)} = \left( e^{(\varepsilon_k - \mu)/T} \pm 1 \right)^{-1}, \qquad (5)$$

where the chemical potential $\mu$ and temperature $T$ are solutions to the system of equations $\sum_k \overline{N_k(E)} = N$ and $\sum_k \varepsilon_k \overline{N_k(E)} = E$. If $\varepsilon_k$ is restricted both from below and above, $T$ can be either positive or negative, corresponding to occupation distributions which decrease or increase, respectively, with the orbital energy. The summation over $k$ in this system can be replaced by integration over the orbital energy. Then $\mu$ and $T$ will depend on the particle density $\tilde{N} = N/L$ and energy density $\tilde{E} = E/L$.

While finding the exact LDOS by direct diagonalization is not possible for large systems, in the case of strong interactions, it can be approximated by the Gaussian shape (see [13])

$$W(E, E_n) \approx C(E) \frac{\Delta_n}{\Delta_\alpha} \exp(-(E - E_n)^2 / \Gamma^2). \qquad (6)$$

where $\Delta_n$ is taken the vicinity of $E_n$ and the normalization factor $C(E)$ is determined by

$$1/C(E) = \int_{E_{\min}}^{E_{\max}} \exp(-(E - E')^2 / \Gamma^2) dE'. \qquad (7)$$

The Gaussian shape approximates the LDOS with good accuracy even for systems of small size [29]. It should be stressed, that the agreement can be provided by the factor $\Delta_n$, which increases LDOS near the energy spectrum boundaries. The resulting distributions, calculated with Eqs. (4), (5), (6), and (7) depend on scaled widths $\tilde{\Gamma} = \Gamma/L$. In addition to the 2D FH and 1D BH model, treated above using exact diagonalization, we consider also the 2D BH model with the same 1B Hamiltonian as the 2D FH one [29]. Figure 2 shows the obtained distributions for the Gaussian width $\Gamma$ covering both eigenstates corresponding to positive and negative temperature, or, respectively, to the decreasing and increasing Gibbs distributions. The resulting distributions for the interacting system are clearly non-monotonic. This effect can not be reduced to a change of temperature. Since $\mathcal{N}_{\mathrm{PC}} \sim \Gamma/\Delta_n$ and $\mathcal{N}_{\mathrm{HS}} \sim (E_{\max} - E_{\min})/\Delta_n$, the ratio of NPC to the Hilbert space dimension can be estimated

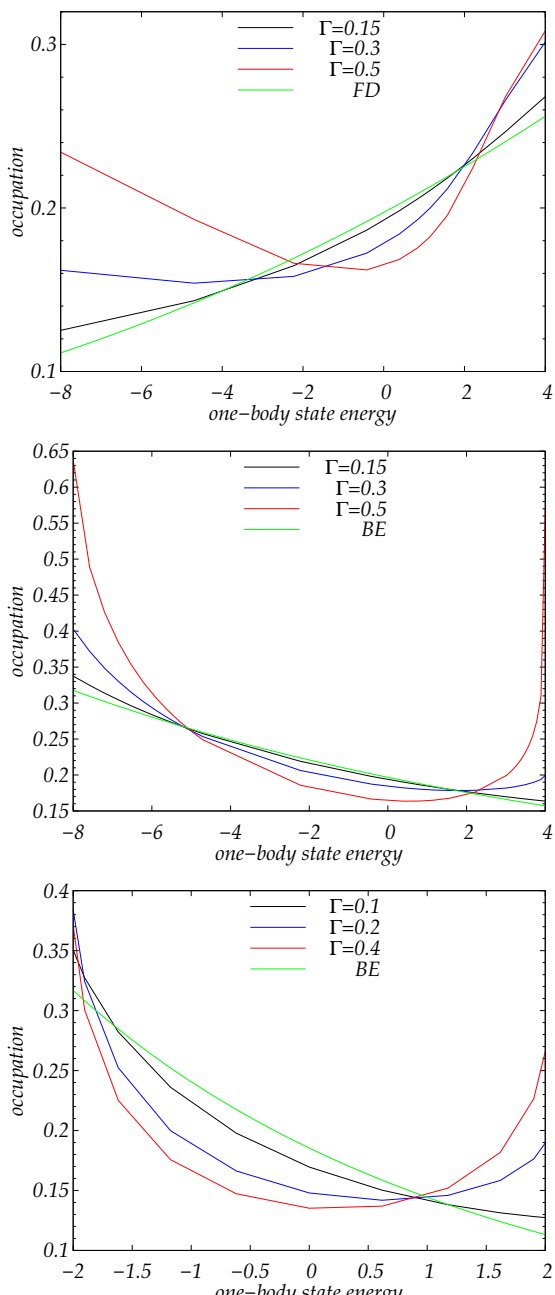

FIG. 2. 1B orbital occupations for different $\tilde{\Gamma}$. (a) The FH model with $\tilde{N} = 0.2$ and $\tilde{E} = 0.1$. (b) The 2D BH model with $\tilde{N} = 0.2$ and $\tilde{E} = -0.1$. (c) The 1D BH model with $\tilde{N} = 0.2$ and $\tilde{E} = -0.1$. The green lines show the Fermi-Dirac or Bose-Einstein distributions corresponding to $\tilde{E}$.

as $\mathcal{N}_{\mathrm{PC}}/\mathcal{N}_{\mathrm{HS}} \sim \Gamma/(E_{\max} - E_{\min}) \approx \tilde{\Gamma}/(\tilde{N}\Delta\varepsilon)$, where the range of $\varepsilon_k$ variation $\Delta\varepsilon$ is 12 for the 2D and 4 for the 1D models. Then, in Fig. 2, $\tilde{\Gamma} = 0.5$ corresponds to $\mathcal{N}_{\mathrm{PC}}/\mathcal{N}_{\mathrm{HS}} \sim 0.2$ for the 2D models and $\tilde{\Gamma} = 0.4$ corresponds to $\mathcal{N}_{\mathrm{PC}}/\mathcal{N}_{\mathrm{HS}} \sim 0.5$ for the 1D BH. These high participation fractions indicate quantum ergodicity [15].

In the MBL literature, the proportionality of NPC to Hilbert space dimension is used as an attribute of localization, distinguishing extended eigenstates from localized eigenstates. This property was reported for eigenstates in Heisenberg [32] and XXZ [33] spin chains, the Jaynes-Cummings-Hubbard system [34], and the Bose-Hubbard model [35]. Then, the ratio $\tilde{\Gamma} \sim \tilde{N}\Delta\varepsilon\mathcal{N}_{\mathrm{PC}}/\mathcal{N}_{\mathrm{HS}}$ should remain unchanged for extended eigenstates in the thermodynamic limit $N \to \infty$, while $\tilde{N} = \mathrm{const}$ and $\tilde{E} = \mathrm{const}$. As a result, the distribution deviations from the Gibbs ones may survive in the thermodynamic limit.

For bosonic systems, there is a clear classical mean-field limit wherein the field operators are replaced by $c$-numbers and their amplitudes and phases serve as conjugate action-angle canonical variables. The observed broadening of the LDOS may then be viewed as resulting from the interaction-induced deformation of the energy shells within the classical phasespace. For the boson models discussed here, there is good quantum-classical correspondence in the sense that mean occupations agree well with semiclassical averages over the pertinent shells (see [28, 36]) and the mean LDOS corresponds to the overlap of the classical shell of the non-interacting system with each of the interacting system's energy shells. While weak interactions only slightly shift the non-interacting shells, strong interactions deform them substantially: The non-interacting shell overlaps with many interacting shells, resulting in the broadening of the LDOS.

Concluding, orbital population distributions in eigenstates of strongly-interacting many-body systems can deviate from the Gibbs distributions while the chaotic nature of eigenstates and their thermalization is confirmed by the energy spectra statistics and by suppression of the eigenstate-to-eigenstate fluctuations of expectation values. This effect can appear when the interactions mix the Gibbs distributions with positive and negative temperatures in lattice systems which — as their energy spectra are restricted both from below and above — allow states with any sign of temperature. The distribution deviations may be observed experimentally with cold atoms in optical lattices.

AV acknowledges support from the NSF through a grant for ITAMP at Harvard University.

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

# SUPPLEMENTAL MATERIAL FOR:
## EIGENSTATE THERMALIZATION TO NON-GIBBS STATES IN STRONGLY-INTERACTING CHAOTIC LATTICE GASES

Numbers of equations and figures in the Supplemental material start with S. References to equations and figures in the Letter do not contain S.

## I. TWO-DIMENSIONAL LATTICE MODELS

The Fermi-Hubbard (FH) model on a two-dimensional (2D) lattice has the Hamiltonian

$$\hat{H}_F = - \sum_{l_x=1}^{L_x} \sum_{l_y=1}^{L_y} \sum_{\delta_x=-1}^{1} \sum_{\delta_y=-1}^{1} \left(1 - \delta_{\delta_x 0}\delta_{\delta_y 0}\right) \hat{a}^\dagger_{l_x l_y} \hat{a}_{l_x+\delta_x l_y+\delta_y} + \hat{V}_F - \bar{V}, \tag{S-1}$$

$$\hat{V}_F = V \sum_{l_x=1}^{L_x} \sum_{l_y=1}^{L_y} \left( \sum_{\delta_x=\pm 1} \hat{a}^\dagger_{l_x l_y} \hat{a}^\dagger_{l_x+\delta_x l_y} \hat{a}_{l_x l_y} \hat{a}_{l_x+\delta_x l_y} + \sum_{\delta_y=\pm 1} \hat{a}^\dagger_{l_x l_y} \hat{a}^\dagger_{l_x l_y+\delta_y} \hat{a}_{l_x l_y} \hat{a}_{l_x l_y+\delta_y} \right), \tag{S-2}$$

where $V$ is the nearest-neighbor interaction strength, the hopping energy is used as the energy unit, $\hat{a}_{l_x l_y}$ are annihilation operators of spin-polarized fermions, and $l_x$, $l_y$ specify location on the the $L_x \times L_y$ lattice. Outside the square $1 \leq l_x \leq L_x$, $1 \leq l_y \leq L_y$, the field operators are defined by the twisted periodic boundary conditions $\hat{a}_{l_x+L_x l_y} = e^{i\chi_x}\hat{a}_{l_x l_y}$, $\hat{a}_{l_x l_y+L_y} = e^{i\chi_y}\hat{a}_{l_x l_y}$. The phase changes $\chi_x = (1+\sqrt{5})/2$ (the golden ratio) and $\chi_y = e/2$ are used in the present calculations. The one-body (1B) orbitals are plane waves with the momentum components

$$p_x = \frac{2\pi m_x + \chi_x}{L_x} \quad (1 \leq m_x \leq L_x), \quad p_y = \frac{2\pi m_y + \chi_y}{L_y} \quad (1 \leq m_y \leq L_y), \tag{S-3}$$

where $m_x$ and $m_y$ are integers. The orbital energies are expressed as

$$\varepsilon_{2D}(p_x, p_y) = -2\cos p_x - 2\cos p_y - 4\cos p_x \cos p_y. \tag{S-4}$$

The $k$th orbital momentum components $p_x(k)$ and $p_y(k)$ are chosen such that the orbitals are labeled in increasing order of their eigenenergies $\varepsilon_k = \varepsilon_{2D}(p_x(k), p_y(k))$. In the limit of the large $L_{x,y}$ the number of the orbitals with energies below $\varepsilon$ can be approximated by

$$\frac{k(\varepsilon)}{L} \approx \frac{1}{(2\pi)^2} \int_0^{2\pi} dp_x \int_0^{2\pi} dp_y \vartheta(\varepsilon - \varepsilon_{2D}(p_x, p_y)), \tag{S-5}$$

where summation over $m_{x,y}$ is approximated by integration over $p_{x,y}$, $L = L_x L_y$ is the total number of orbitals, and $\vartheta$ is the Heaviside step function. Inversion of $k(\varepsilon)$ allows us to express $\varepsilon_k = \varepsilon(k/L)$ in terms of lattice-size independent function $\varepsilon(\tilde{k})$ which increases with $\tilde{k}$ from $\varepsilon(0) = -8$ to $\varepsilon(1) = 4$ (see Fig. S1).

The eigenstates of the Hamiltonian (S-1) with $V = 0$ are thus the orbital Fock states $|n\rangle = |n_1, ... n_L\rangle$ where $0 \leq n_k \leq 1$ is the integer occupation of the $k$-th orbital, and $\sum_{k=1}^{L} n_k = N$. Due to spatial homogeneity of the Hamiltonian (S-1), we consider separately each segment with given total momentum components $P_x$ and $P_y$, such that $\sum_{k=1}^{L} n_k p_{x,y}(k) = P_{x,y}$. The orbital Fock states for each segment constitute a $\mathcal{N}_{HS} \approx (L-1)!/(N!(L-N)!)$ dimensional complete basis for the many-body Hilbert space with given $P_x$ and $P_y$. Representing the full Hamiltonian in this basis and diagonalizing, we obtain the exact many-fermion eigenstates $|\alpha\rangle$.

The average expectation values of interactions $\bar{V} = \sum_\alpha \left\langle \alpha \left| \hat{V}_F \right| \alpha \right\rangle / \mathcal{N}_{HS}$ is subtracted in the Hamiltonian (S-1) in order to provide a substantial overlap between the non-interacting and interacting spectra $\{E_n\}$ and $\{E_\alpha\}$. Due to completeness of the set $|\alpha\rangle$, we have

$$\bar{V} = \frac{1}{\mathcal{N}_{HS}} \sum_n \left\langle n \left| \hat{V}_F \right| n \right\rangle, \tag{S-6}$$

where diagonal matrix elements of the interaction (S-2) can be expressed as

$$\left\langle n \left| \hat{V}_F \right| n \right\rangle = \frac{4V}{L} \sum_{k<k'} n_k n'_k \left( \sin^2 \frac{p_x(k) - p_x(k')}{2} + \sin^2 \frac{p_y(k) - p_y(k')}{2} \right). \tag{S-7}$$

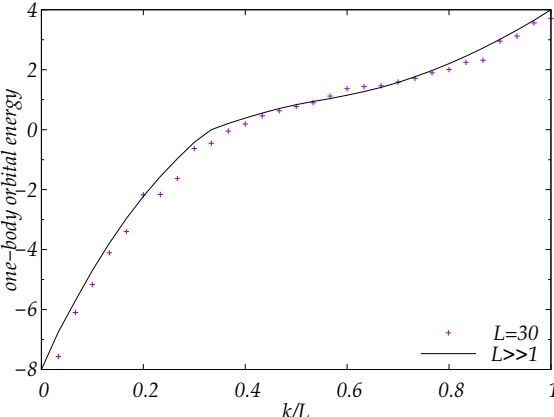

FIG. S1. One-body orbital energy as a function of the orbital label $k$ for small and large FH models.

As all orbitals are presented unbiasedly in the set $\{|n\rangle\}$, we can approximate the average over the Hilbert space in (S-6) by the average over $p_x(k)$ and $p_y(k)$, i.e, replace squared sines in (S-7) by $1/2$. As a result, we get

$$\bar{V} \approx 2N(N-1)\frac{V}{L}. \tag{S-8}$$

This approximate value will be valid as well for the average over each microcanonical interval, where the orbitals are presented unbiasedly. The stretching of the interacting spectrum $\{E_\alpha\}$ in comparison with the non-interacting one $\{E_n\}$ is related to the level repulsion, which is beyond the first order effect in $\hat{V}_F$.

The expectation values of the orbital occupations $\langle\alpha|\hat{N}_k|\alpha\rangle$ for each of the $\mathcal{N}_{\mathrm{HS}}$ eigenvalues are calculated with $P_x = 3$, $P_y = 2$, and $\mathcal{N}_{\mathrm{HS}} = 19448$ for $N = 6$ particles in $L = 30$ sites of the $6 \times 5$ lattice.

We also consider a 2D Bose-Hubbard (BH) model of the large system size. It has the Hamiltonian (S-1) where $\hat{a}_{l_x l_y}$ are annihilation operators of spinless bosons and $\hat{V}_F$ is replaced by local interactions. The 1B Hamiltonian, orbitals, and $\varepsilon_k$ for this model are the same as for the 2D FH one.

## II.   ONE-DIMENSIONAL BOSE-HUBBARD MODEL

The tight binding bosonic Hamiltonian on a one-dimensional (1D) lattice (in units of the hopping rate) reads,

$$\hat{H}_B = -\sum_{l,m=1}^{L} \hat{b}_l^\dagger J_{lm} \hat{b}_m + \frac{1}{2} V \sum_{l=1}^{L} \hat{n}_l(\hat{n}_l - 1) - \bar{V}, \tag{S-9}$$

where $l = 1, ..., L$ is the site index, $J_{lm} = J_{ml}^*$ is the hopping matrix coupling sites $l$ and $m$, $V$ is the on-site interaction strength, $\hat{n}_l = \hat{b}_l^\dagger \hat{b}_l$ is the number of bosons at site $l$, and $b_l$ are bosonic particle annihilation operators. Throughout the manuscript we have used the Bose-Hubbard (BH) configuration $J_{l \neq m} = \delta_{l,m \pm 1}$ with hard wall boundaries, i.e. a linear chain of $L$ sites. For this configuration, the dynamical behavior of the system, e.g. its degree of chaoticity, is set by the dimensionless interaction parameter $u = VN$. In order to remove the remaining parity symmetry and increase chaoticity, we have introduced a weak random 'disorder' on-site potential $J_{l,l} = \mathrm{rnd}[-0.05, 0.05]$.

The 1B orbitals are found by diagonalizing the hopping matrix, thereby obtaining the eigenvectors $\{f_\alpha\}_{k=1,...L}$ and the orbital energies $\varepsilon_k$. Defining the bosonic mode annihilation operators $\hat{c}_k = \sum_l f_k(l) \hat{b}_l$ where $f_k(l)$ denotes the $l$-th component of the $k$-th eigenvector, we obtain the orbital number operators:

$$\hat{N}_k = \hat{c}_k^\dagger \hat{c}_k = \sum_{l,m} f_k^*(l) f_k(m) \hat{b}_l^\dagger \hat{b}_m \tag{S-10}$$

The BH Hamiltonian then transforms in the orbital basis into,

$$\hat{H}_B = \sum_{k=1}^{L} \varepsilon_k \hat{c}_l^\dagger \hat{c}_l + \hat{V}_B - \bar{V}, \tag{S-11}$$

where,

$$\hat{V}_B = \sum_{k,k',k'',k'''=1}^{L} u_{k,k',k'',k'''} \hat{c}_k^\dagger \hat{c}_{k'}^\dagger \hat{c}_{k''} \hat{c}_{k'''} \tag{S-12}$$

and

$$u_{k,k',k'',k'''} = \frac{V}{2} \sum_{i=1}^{L} f_k^*(i) f_{k'}^*(i) f_{k''}(i) f_{k'''}(i) \tag{S-13}$$

Note that in contrast to the FH model of the previous section, the system is not translationally invariant. Hence there is no momentum conservation law that reduces the allowed four-wave-mixing transitions induced by the interactions between the orbitals.

The eigenstates of the Hamiltonian of Eq. (S-11) with $V = 0$ are thus the orbital Fock states $|n\rangle = |n_1, ... n_L\rangle$ where $n_k$ is the integer occupation of the $k$-th orbital, and $\sum_{k=1}^{L} n_k = N$. The orbital Fock states constitute a $\mathcal{N}_{\mathrm{HS}} = (N + L - 1)!/(N!(L - 1)!$ dimensional complete basis for the many-body Hilbert space (throughout the manuscript $\mathcal{N}_{\mathrm{HS}} = 19448$ for $N = 10$ particles in $L = 8$ sites). Representing the full Hamiltonian in this basis and diagonalizing, we obtain the exact many-boson eigenstates $|\alpha\rangle$ and calculate the expectation values of the orbital occupations $\langle\alpha|\hat{N}_k|\alpha\rangle$ for each of the $\mathcal{N}_{\mathrm{HS}}$ eigenvalues. In this model, $\bar{V} = \overline{\left\langle\alpha\left|\hat{V}_B\right|\alpha\right\rangle}$ is the microcanonical mean of the interaction expectation value. For bosons, due to multiple orbital occupations, $\bar{V}$ is energy-dependent. Then, it is numerically calculated for each microcanonical shell.

## III. CHAOTIC PROPERTIES

The degree of chaoticity of a quantum system can be deduced from its level spacing statistics. One measure of the transition from the Poissonian statistics of integrable systems to the Wigner-Dyson statistics of completely chaotic systems is the ratio of consecutive level spacings [30]

$$r_\alpha = \frac{\min(E_{\alpha+1} - E_\alpha, E_\alpha - E_{\alpha-1})}{\max(E_{\alpha+1} - E_\alpha, E_\alpha - E_{\alpha-1})}. \tag{S-14}$$

averaged over the entire spectrum or over a pertinent energy shell. The value $\langle r\rangle = 2\ln 2 - 1 \approx 0.38629$ is indicative of Poissonian statistics, whereas $\langle r\rangle = 4 - 2\sqrt{3} \approx 0.53590$ is obtained for Wigner-Dyson GOE statistics [31]. In Fig. S2 we present this measure as a function of the interaction strength for our model systems. The 1D BH system is integrable at weak interaction due to its near-separability and at strong interaction due to macroscopic self-trapping where site occupations become integrals of motion. By contrast, the 2D FH system does not return to integrability at high interaction strength.

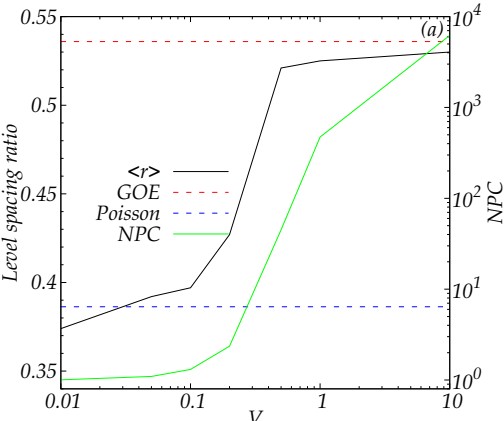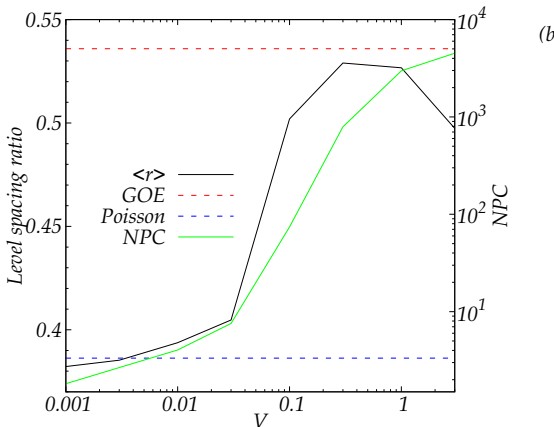

FIG. S2. (a) Level spacing ratio (black line) vs. interaction strength for the FH model. Dashed lines show $\langle r\rangle$ for the Poisson and GOE statistics. The green line shows NPC. (b) The same for the 1D BH model.

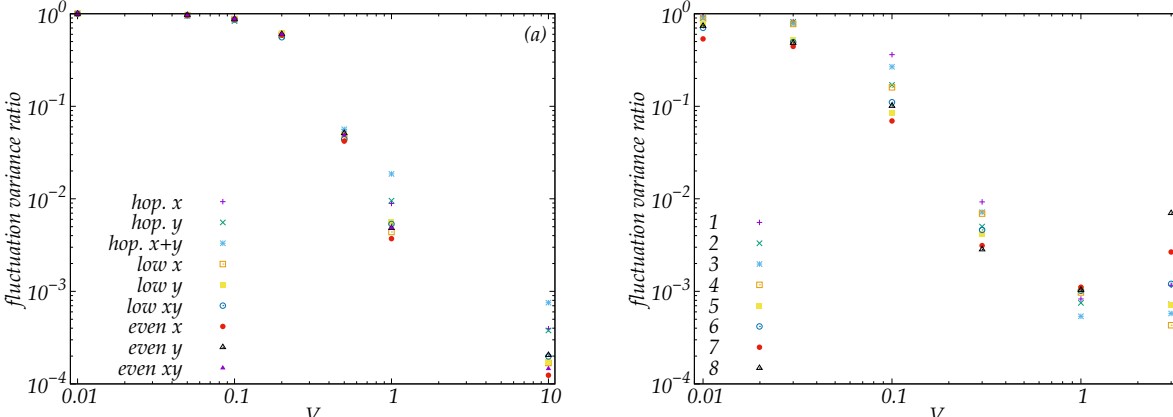

FIG. S3. Ratio of eigenstate-to-eigenstate fluctuation variances for the non-integrable to ones for the integrable systems eigenstates vs. interaction strength for: (a) the FH model averaged over the microcanonical shell with the mean energy 0; (b) the 1D BH model with the mean shell energy $-5.14$.

Chaos can also be characterized by the NPC (see Fig. S2) and the eigenstate-to-eigenstate fluctuations of the observable expectation values (see Fig. S3). Note that a high NPC is a necessary but not sufficient condition for chaos, as the number of eigenstates of a non-interacting system participating in an eigenstate of the interacting system can be large even if the latter is integrable. Due to the large number of the orbitals in the FH model, we consider cumulative observables: the total occupations of orbitals with $m_x < L_x/2$ and any $m_y$ in Eq. (S-3) [low x in Fig. S3(a)], with $m_y < L_y/2$ and any $m_x$ (low y), with $m_x < L_x/2$ and $m_y < L_y/2$ (low xy), with even $m_x$ and any $m_y$ (even x), with even $m_y$ and any $m_x$ (even y), and with even $m_x$ and $m_y$ (even xy). We also consider the hopping energies in the $x$

$$-\sum_{l_x=1}^{L_x}\sum_{l_y=1}^{L_y}\sum_{\delta_x=\pm 1}\hat{a}^\dagger_{l_x l_y}\hat{a}_{l_x+\delta_x l_y}$$

and $y$

$$-\sum_{l_x=1}^{L_x}\sum_{l_y=1}^{L_y}\sum_{\delta_y=\pm 1}\hat{a}^\dagger_{l_x l_y}\hat{a}_{l_x l_y+\delta_y}$$

directions, as well as the sum of these energies. For the 1D BH model, due to the small number of orbitals, we consider the individual orbital occupations.

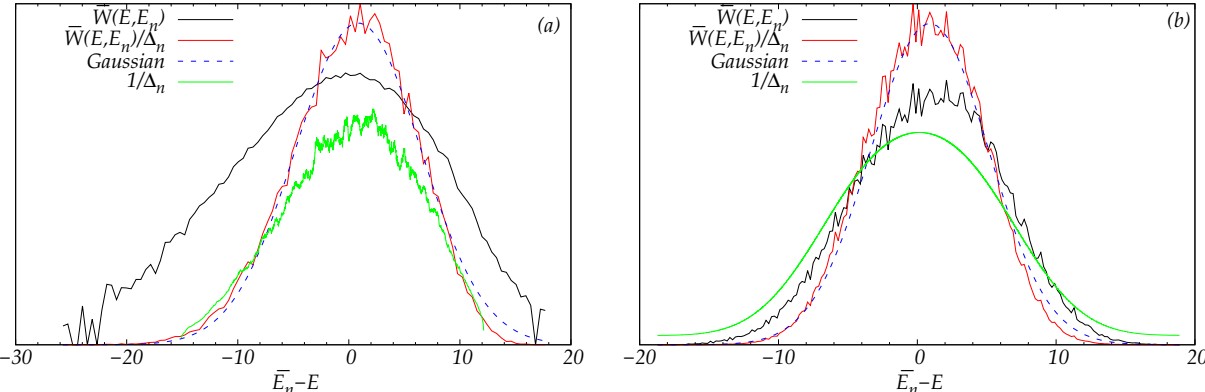

FIG. S4. Local density of states, averaged over the orbitals for: (a) the FH model with $V = 10$ at $E = -5.2$; (b) the 1D BH model with $V = 3$ at $E = -5.14$.

## IV.   LOCAL DENSITY OF STATES

Figure (S4) demonstrates the local density of states (LDOS) [see Eq. (3)] averaged over the orbitals

$$\bar{W}(E, \bar{E}_n) = \frac{\Delta_n}{\Delta_{\mathrm{MC}}} \sum_{n \in \mathrm{MC}(\bar{E}_n)} W(E, E_n)$$

and the averaged LDOS divided by $\Delta_n$ in a comparison with the Gaussian profiles.

For the 1D BH model both $\bar{W}(E, \bar{E}_n)$ and $\bar{W}(E, \bar{E}_n)/\Delta_n$ can be approximated by Gaussian profiles since the density of states $1/\Delta_n$ has a Gaussian shape too. However, in the case of the FH model, the wings of density of states drop abruptly, and only $\bar{W}(E, \bar{E}_n)/\Delta_n$ has a Gaussian shape, but $\bar{W}(E, \bar{E}_n)$ does not.