# Peer review of "Eigenstate thermalization to non-monotonic distributions in strongly-interacting chaotic lattice gases"

_SciPost Physics_

## Round 2 · Referee Report · Anonymous (Referee 1) · 2025-8-11

Strengths

  1. The authors present numerical investigations of quantum strongly-interacting many-body systems. By direct numerical diagonalization of the Hamiltonian they are able to show that these systems, for sufficiently strong coupling depart from their reference single-body statistical distribution (i.e., Fermi-Dirac or Bose-Einstein).
  2. The authors argue that the reason for this discrepancy is due to an effect of mixing independent microcanonical distribution, in a way affine to the mechanisms theorized for the generation of superstatistical distributions [see, e.g.,, C. Beck and E.G.D. Cohen, "Superstatistics", Phys. A, 322, 267275 (2003)] or [S. Davis, "Fluctuating temperature outside superstatistics: Thermodynamics of small systems", Phys. A, 589, 126665 (2022)].
  3. On the basis of the theory argued in point (2), the authors predict the possibility of non-monotonic distributions as a function of the energy: i.e. a mixture of positive and negative temperature distribution. This prediction is novel, as far as I know.

Weaknesses

  1. The explicit reference to the superstatistical literature, which basically predates the theory advanced in this work, is lacking.
  2. The interpretation of the numerical results advanced by the authors is suggestive and plausible but, at this stage, is just a speculation, not yet supported by the authors' numerical studies. The authors argue--reasonably--that the kernel' width \Gamma scales with the interaction strength V yet, at the largest values for V employed in the numerical simulations, the resulting distributions in Fig. 1 do not bear particular quantitative resemblances to those produced in Fig. 2.

Report

I believe that the paper deserves publication but the authors should cure the two weak point mentioned above, namely: 1. Make clear the connection between their theory and the existing literature about non-standard statistics. 2. Make clearer the speculative character of their theory, at this stage.

Requested changes

The change requested are indicated under the "weaknesses" and "report" sections.

Recommendation

Ask for minor revision

  • validity: good
  • significance: good
  • originality: high
  • clarity: good
  • formatting: excellent
  • grammar: perfect

Author:  Vladimir Yurovsky  on 2025-09-30  [id 5875]

(in reply to Report 1 on 2025-08-11)

We thank the referee for careful reading of the manuscript and for recommending publication after minor revision. We have followed his recommendation and revised the ms accordingly. Below is a detailed reply to the referee's report:

Strengths

The authors present numerical investigations of quantum strongly-interacting many-body systems. By direct numerical diagonalization of the Hamiltonian they are able to show that these systems, for sufficiently strong coupling depart from their reference single-body statistical distribution (i.e., Fermi-Dirac or Bose-Einstein). The authors argue that the reason for this discrepancy is due to an effect of mixing independent microcanonical distribution, in a way affine to the mechanisms theorized for the generation of superstatistical distributions [see, e.g.,, C. Beck and E.G.D. Cohen, "Superstatistics", Phys. A, 322, 267275 (2003)] or [S. Davis, "Fluctuating temperature outside superstatistics: Thermodynamics of small systems", Phys. A, 589, 126665 (2022)].

The connection to superstatistics is discussed (see list of changes).

On the basis of the theory argued in point (2), the authors predict the possibility of non-monotonic distributions as a function of the energy: i.e. a mixture of positive and negative temperature distribution. This prediction is novel, as far as I know.

Weaknesses

The explicit reference to the superstatistical literature, which basically predates the theory advanced in this work, is lacking.

The references are cited in the revised manuscript (see list of changes).

The interpretation of the numerical results advanced by the authors is suggestive and plausible but, at this stage, is just a speculation, not yet supported by the authors' numerical studies. The authors argue--reasonably--that the kernel' width \Gamma scales with the interaction strength V yet, at the largest values for V employed in the numerical simulations, the resulting distributions in Fig. 1 do not bear particular quantitative resemblances to those produced in Fig. 2.

The numerical results at the largest value of V in Fig. 1 demonstrate the deviation of the occupation distribution from the microcanonical expectation for mesoscopic systems away from the thermodynamic limit. As shown in [28], even these single-shell microcanonical distributions are different from the Fermi-Dirac or Bose-Einstein distributions that are obtained in the thermodynamic limit. Thus, while Fig. 2 demonstrates the interaction-induced mixing of FD or BE distributions obtained for the microcanonical shells of a macroscopic system (ie in the thermodynamic limit), Fig. 1 shows the mixing of shell-distributions that do not follow FD or BE. The difference in the resulting strong-interaction distribution is hence expected. We have stressed this in the revised manuscript, clarifying that our findings are not restricted to the mixing of large-system BE or FD distributions, but extends also to the non-BE/FD mesoscopic regime [e.g. Ref. [28]).

Report

I believe that the paper deserves publication but the authors should cure the two weak point mentioned above, namely: 1. Make clear the connection between their theory and the existing literature about non-standard statistics. 2. Make clearer the speculative character of their theory, at this stage.

In the revised manuscript, we make clear the connection to non-standard statistics and additionally clarify the difference between the small-system distributions obtained by direct diagonalization in Figure. 1 and the non-monotonic large-system distributions in Fig. 2. We think the latter could not really be characterized as 'speculative', as the only assumptions involved in their derivation are the ubiquitous applicability of Fermi-Dirac or Bose-Einstein occupation distributions for the microcanonical shells of large non-interacting systems, and the well-established broadening of the LDOS at strong interactions. These are the only ingredients required to obtain non-monotonic distributions as in Figure. 2, and while the former ingredient (BE/FD mean occupations in a microcanonical shell) is missing in the case of small systems we are confident it will prevail as the number of particles and degrees of freedom is increased.

Requested changes

The change requested are indicated under the "weaknesses" and "report" sections.

The requested changes are made (see above)

Recommendation

Ask for minor revision

---

## Round 2 · Referee Report · Anonymous (Referee 2) · 2025-8-28

Report

The full report is contained in the attached file "ReportSciPost.pdf"

Attachment

Recommendation

Reject

  • validity: ok
  • significance: low
  • originality: low
  • clarity: low
  • formatting: -
  • grammar: reasonable

Author:  Vladimir Yurovsky  on 2025-09-05  [id 5783]

(in reply to Report 2 on 2025-08-28)
Category:
answer to question
reply to objection

Dear Editor, I read carefully the manuscript “Eigenstate Thermalization to non-monotonic distributions in strongly interacting chaotic lattice lattice gases”, submitted for publication to SciPost Physics and, to be honest, I do not find it very convincing, neither very well explained and motivated.

We thank the referee for careful reading.

The authors consider interacting bosons (1d) and fermions (2d) systems, studying the occupancy of single particle orbitals. Their main finding is that, even when increasing the number of degrees of freedom, remarkable deviations from the Bose-Einstein or Fermi-Dirac statistics for the single particle levels are found. My main question/concern is the following: why we should expect even for a strongly interacting system of quantum particles that the free particles statistics should be followed for the occupancy of single particle states? Perhaps the authors follows a (common?) logic according to which a strongly interacting quantum system can be seen as a system of “free” single particle states in contact with a thermal bath, where the role of the thermal bath is provided by the interaction part of the Hamiltonian?

A substantial body of work exists on the thermalization of interacting systems, including the Bose-Hubbard model under study (see [2,8-10] in the manuscript). Weak interactions play the role of collisions in the Boltzmann standard picture, inducing transitions between the natural modes of the non-interacting system. The mere deviations from Bose-Hubbard or Fermi-Dirac statistics at strong interaction is indeed not surprising. However, the main result of our work isn’t their existence. It is their specific form, i.e. their non-monotonic nature which is starkly different from the canonical forms. This is stated clearly in the manuscript, starting from its title.

This is sometimes a “bona fide” assumption which works classical systems, supported by numerical evidence, but it is rather a speciial case rather than a general fact to be expected. I do not see why things should work different for quantum ones. In case, please explain me.

We do not make the assumption the referee alludes to.

First of all the authors should clarify much better the context and provide in the introduction the general and strong examples needed to convince the reader that also for strongly interacting systems the quantum free particles statistics are to be expected for the single-particle levels of the interacting ones.

References on thermalization in interacting systems are provided in [2,8-10], and discussed in the introduction of our manuscript. In contrast to weakly interacting systems, the claim that free particle statistics applies to strongly interacting systems is never made throughout the manuscript. As above, our main finding is the non-monotonic nature of the effective quasimomentum distribution which is of value to experimentalists in the field.

I would have been more convinced by an argument proposing that weakly interacting quasi- particles excitation follows free particles statistics, but this does seems the case, all their core arguments are not build on quasi-particles modes of their systems. What are the cases and the reasons why single particle levels of interacting systems should obey independent particles statistics?

It is indeed plausible to assume that quasiparticle distributions follow BE or FD. However, this is not what we do here. Calculating quasiparticle populations is subtle, as unlike the natural modes of the non-interacting system, quasiparticle modes do not constitute a complete basis set for the one-particle phasespace. Our motivation to focus on the natural modes of the non-interacting system is prosaic: quasimomenta are often measured in experiments (e.g. by quenching down the interactions) and their distribution is of practical interest, even for strongly interacting systems. As the referee points out, there is no reason to expect the quasimomenta distributions will follow BE or FD in this case. However, the fact that they are non monotonic in systems whose temperature varies from positive to negative, is worth highlighting.

Then, I am not covinced and I am not familiar with the description of the Eigenstate Thermalization Hypothesis (ETH) at the beginning of paper, in particular Eq. (1), which is only saying that the expectation value of the observable \hat{O} varies across neighbouring eigenstates of the full interacting Hamiltonian smoothly enough that its expectation on a single eigenstate can be replaced with its average over neighbouring ones. This is only part of the ETH, which also requires a “strong” assumption on the exponential decay with system size of the off-diagonal matrix elements of the kind Hp. 2 <\alpha | \hat{O} | \beta > ~ exp{- S(E_av)} where E_av = (E_\alpha + E_\beta) / 2 and S(E_av) is the corresponding entropy, where clearly it is assumed that the entropy is extensive. This second hypothesis is crucial to guarantee thermalization and “independence” on initial conditions. Indeed the more correct enunciation of ETH is that by choosing any initial state | \psi_0 > the time evolution with Hamiltonian \hat{H} of this state induces for almost all observables \hat{O} and almost all times a probability distribution on the measures of \hat{O}which cannot be distinguished from the microcanonical one on the energy shell [E – DE , E + DE], where E = < \psi_0 | \hat{H} | psi_0 >. Something rather different from what written by the authors in the introduction and where the role of dynamics and of initial conditions is clear (which is not in the present discussion). This point should be clarified.

The dynamical statement made by the referee is one-to-one equivalent to the statement that the phasespace distributions of all eigenstates within the shell are smeared over it, hence the expectation value of most local observables, evaluated over any eigenstate within the shell is equal to its microcanonical mean. Eq. (1) states that the expectation value of any local observable \hat{O}, evaluated with any eigenstate |alpha> in a microcanonical shell is equal to a microcanonical mean over all |alpha>. This automatically implies that all expectation values within the shell are equal. Indeed, the only way for all arbitrary local operators to have nearly the same value for all states within a microcanonical shell is if the eigenstates are thermalized and their LDOS differ only statistically.

Starting from this initial flaw in the presentation of ETH, I find the whole following discussion and analysis of the chaoticity properties of the system not very convincing. Can we really trust simply the average value of eigenvalues spacings as a reliable order parameter from the transition from “near integrability” to “chaos”? I general the statement on chaoticity or integrability is about the full distribution of eigenvalues spacings. In full generality, it is really hard to tell the difference between two probability distributions only from the knowledge of their first moment.

The referee seems to be unfamiliar with the mean adjacent-level spacing-ratio criterion. The parameter <r> is not mean eigenvalue spacing but the mean spacing-ratio which (like e.g. the Brody parameter) is indicative of the transition from Poisson to Wigner-Dyson statistics. This criterion has been introduced in [29], and is very widely used as a clear evidence of integrability-chaos transition (1600 citations of [29] attest to its validity). Needless to say we have long ascertained Wigner-Dyson level spacing statistics independently for the systems under study, see e.g. Ref. [28] and references within.

For instance, with reference to Fig. 4 in the appendices, if one also looks at the distributions, does he also find a consistent transition from a Poissonian-like shape to a Wigner-Dyson shape?

Yes. This was shown to be the case in many hundreds works on dozens of systems since its first publications [29]. For the specific BH system under study, see e.g. [28] where we also calculate the Kullback-Leibler divergence of the level spacing distribution from Wigner-Dyson and the deviation from the (known, see there) r-distribution. All three measures expectedly agree.

The authors should also be much more clear about the definition of principal components, only presented at the beginning as “the number of integrable system eigenstates comprising the non- integrable ones”. This definition seems rather obscure, just a jergon used by people working daily on this topic. What does it means? Is it a sort of statement on the “vicinity” between integrable and non- integrable systems eigenstates? Or is a statement on how many integrable eigenvalues fills the gaps between non-integrable eigenvalues? And why is this related to ETH and chaoticity?

The eigenstates of the non-integrable system are superpositions of the integrable-system eigenstates. The participation number (or 'Number of Principal Components') simply estimates (for each exact eigenstate) the number of contributing integrable-system eigenstates. This standard definition will be clarified in the text.

Let me now jump to the core section of the paper, which is Sec. 4, “Non-monotonic distributions”. Again, I find the presentation really not clear. First of all, as a general impression, it seems to me that only here is presented a a “rough” argument to demonstrate what it is on the contrary assumed by the very beginning as a general fact, namely that the statistics of single particle levels occupancy should be similar in interacting a non interacting systems. Then, notations and explanations in my opinion are not clear. At a certain point it appears a number operator \hat{N}_k where it is not specified what the subscript index “k” refers to. They say “Consider a particular case of the orbital occupation operator…”. Which orbital? The single-particle non interacting system orbital? So, in this case, what the index “k” referes to? Then, again without a definition is introduced “\overline{N}_k(E)” and is declared that the “shape of the microcanonical distribution of orbital occupation”--- which has now become the “\overline{N}_k(E)”--- “alters”. Alters with respect to what? With respect to which previous form? Then in the following line, in a chain of equalities which should be the core argument to say that the single-particles levels statistics is similar in interacting and non-interacting systems, it comes out of the blue a new symbol, “\overline{N}_k^{int}(E)”, which makes me even more confused about the meaning of its previous version without the superscript “^{int}”. All these inaccuracies in the presentation makes really difficult to appreciate the relevance and rigour of the whole discussion, the core argument and the main findings.

The presentation will be improved.

Let me now mention another emphatic assertion of the authors, which to me seems a very general and trivial fact and, honestly, they have not succeeded in convincing me that it is not: “Our main point is that the interactions mix different microcanonical energy shells of the non- interacting system, so that the microcanonical occupation means over the interacting system’s energy shell do not match any of the corresponding microcanonical means over non interacting shells.”

This assertion is taken by the referee out of context of the discussion of the previous work [10], which reduces the effect of interaction to the energy shift, leading to the Bose-Einstein distribution with modified parameters. The assertion is necessary to explain why the prescription [10] doesn’t work in the case of strong interactions.

In its present form, I suggest rejectance of this manuscript. Due to either a lack of strength of the results or a lack of accuracy in the presentation, I do not see any true novel physical message or finding beside the almost trivial observation that the properties of an interacting and non-integrable quantum system are remarkably different from that of free quantum particles or from that of an integrable system.

The main result of the present work --- the non-monotonic quasimomenta distributions in strongly-chaotic systems --- is ignored by the referee, although it is highlighted in the title. We fully agree with the referee that deviations of these distributions from BH and FD are to be expected. However, the non-monotonic form of these practically relevant distributions is noteworthy and was never discussed in the literature. We think this is sufficiently novel to justify publication.

---

## Round 3 · Referee Report · Anonymous (Referee 2) · 2025-10-29

Report

I acknowledge the authors made the amendments and additions to the text which I asked them in order to improve clarity, for which I thank them. Still, they did not succeeded in convincing me about the overall significance/impact of their results. In my opinion the paper is definitely targeted to a very specialized audience, the only which can fully appreciate all the details of their analysis and the implication of their results. For this reason I think the paper is definitely more adapt for "SciPost Physics Core" rather than "SciPost".

Let me add some comments to further clarify my opinion.

The authors strongly emphasize the finding of non-monotonic deviations
from the Bose-Einstein and Fermi-Dirac statistics. While they agree with me
that a deviation from the free-particle statistics is not surprising in itself for
a strongly interacting system, they stress the peculariarity of the "non-monotonicity". But what does this "non-monotonicity" means?
Which is the peculiarity of the system revealed by the non-monotonicity? What do we learn/understand from it? Which is the difference between interacting systems with "monotonous" and "non-monotonous" deviations from the free-particle statistics?

Statements such us "Our main point is that the interactions mix different microcanonical shells of the non-interacting system, so that the microcanonical occupation means over the interacting system’s energy shell do not match any of the corresponding microcanonical means over non-interacting shells" seem indeed quite trivial from a general perspective, neither particularly deep or particularly profound.

Also the final sentence

"Unlike previously observed non-monotonic occupa-
tion distributions in weakly-interacting mesoscopic systems [28], this strong-interaction effect appears due to the mixing of microcanonical shells with temperatures of opposite sign and survives in large systems. The distribution deviations may be observed experimentally with cold atoms in optical lattices."
is not very convincing about the general relevance of the results. An explanation
of the relation between the mixing of microcanonical shells with different temperatures and the presence of non-monotonic nature of deviations from free particle statistics should be at least attempted, more convincing and extended arguments presented to convince a general audience to the meaning/relevance of this finding. Last, but not least, the connection drawn in the last sentence with possible experiments is definitely too vague.

Recommendation

Accept in alternative Journal (see Report)

---

## Round 3 · Referee Report · Anonymous (Referee 1) · 2025-10-29

Strengths

The strengths of the work have been listed in my previous report. I am goingo to repeat them here below. 1. The authors present numerical investigations of quantum strongly-interacting many-body systems. By direct numerical diagonalization of the Hamiltonian they are able to show that these systems, for sufficiently strong coupling depart from their reference single-body statistical distribution (i.e., Fermi-Dirac or Bose-Einstein). 2. The authors argue that the reason for this discrepancy is due to an effect of mixing independent microcanonical distribution, in a way affine to the mechanisms theorized for the generation of superstatistical distributions [see, e.g.,, C. Beck and E.G.D. Cohen, "Superstatistics", Phys. A, 322, 267275 (2003)] or [S. Davis, "Fluctuating temperature outside superstatistics: Thermodynamics of small systems", Phys. A, 589, 126665 (2022)]. 3. On the basis of the theory argued in point (2), the authors predict the possibility of non-monotonic distributions as a function of the energy: i.e. a mixture of positive and negative temperature distribution. This prediction is novel, as far as I know.

Weaknesses

I had pointed to two weaknesses in my previous report.

  1. The explicit reference to the superstatistical literature, which basically predates the theory advanced in this work, is lacking.
  2. The interpretation of the numerical results advanced by the authors is suggestive and plausible but, at this stage, is just a speculation.

I feel that the authors have satisfactorily replied to these issues in their revised version

Report

In my opinion, the manuscript meets the acceptance criteria.

Recommendation

Publish (meets expectations and criteria for this Journal)

---

## Round 3 · Author Response

Dear Editor,
We thank the referees for careful reading. The manuscript is modified according to their suggestions and comments.

---

## Round 3 · List of Changes

In the beginning of the 4th paragraph in introduction, the definition of the number of principal components is clarified, in response to referee 2.

The 2nd paragraph in Sec. 4 (started from “Consider”) is extended and definitions therein are clarified, in response to referee 2.

Further discussion of the difference between the mesoscopic systems Fig. 1 and the macroscopic systems Fig. 2 has been added to Sec. 4 and the beginning of Sec. 5, in response to the question raised by referee 1.

In the end of Sec. 5 we discuss relation to superstatistics and cite the references suggested by referee 1.

In the beginning of App. C, the discussion of the level spacing ratio is extended, in response to referee 2.

---

## Editorial Decision

in_voting